# New Isolates of Betachloroviruses Shed Light on the Diversity and Biological Complexity of an Unexplored Group of Giant Algal Viruses

**DOI:** 10.3390/v17081096

**Published:** 2025-08-08

**Authors:** Júlia W. Souza, Lethícia R. Henriques, Roger M. Carlson, Bruna B. F. Botelho, João Victor R. P. Carvalho, João Pedro N. Santos, Eric R. G. R. Aguiar, Irina V. Agarkova, James L. Van Etten, David D. Dunigan, Rodrigo A. L. Rodrigues

**Affiliations:** 1Laboratório de Vírus, Departamento de Microbiologia, Universidade Federal de Minas Gerais, Belo Horizonte 31270-901, MG, Brazil; 2Núcleo de Apoio Técnico ao Ensino, Pesquisa e Extensão, Instituto de Ciências Ambientais, Químicas e Farmacêuticas, Universidade Federal de São Paulo, Diadema 09913-030, SP, Brazil; 3Nebraska Center for Virology, University of Nebraska, Lincoln, NE 68583-0900, USA; 4Department of Plant Pathology, University of Nebraska, Lincoln, NE 68583-0722, USA; 5Laboratório de Bioinformática de Vírus, Universidade Estadual de Santa Cruz, Ilhéus 45662-900, BA, Brazil

**Keywords:** *Phycodnaviridae*, *Chlorovirus*, *Micractinium conductrix*, viral diversity, genomovars, plaque phenotype

## Abstract

The majority of giant algal viruses belong to the family *Phycodnaviridae*, class *Algavirales*, phylum *Nucleocytoviricota*. Among them, the genus *Chlorovirus* is the most studied, with three recognized groups based on genomics and host range, although many fundamental questions remain to be elucidated, particularly regarding their diversity. In this study, we focus on betachloroviruses, a poorly explored subgroup that infects the alga *Micractinium conductrix* Pbi. Here, we describe the isolation and genomic analysis of 11 new betachloroviruses from water samples collected in Nebraska, USA. With 25 fully sequenced genomes now available, we assessed the genomic diversity of these viruses. They have double-stranded DNA genomes ranging from 295 to 374 kbp, encoding hundreds of ORFs, of which a large number (~40%) lack known function. Comparative genomics and phylogenetic analyses revealed three species of betachlorovirus, each with high intra-species genomic identity. Notably, some isolates with over 99.5% genomic identity display markedly different plaque phenotypes, which led us to propose the use of the term genomovar among giant algal viruses, a concept potentially applicable to other giant viral groups yet to be explored. Altogether, this work advances our understanding of betachloroviruses and highlights the importance of linking viral genotype to phenotype, opening new avenues for exploring the diversity of giant algal viruses.

## 1. Introduction

Aquatic environments are rich in biodiversity and harbor an astronomical number of viruses, with estimates indicating around 1.8 × 10^27^ viral particles in freshwater environments [1]. Most of these viruses infect prokaryotes, but a significant portion infect diverse eukaryotes, from large vertebrates to the smallest unicellular organisms, such as amoebas and algae. These viruses include the so-called giant viruses, officially classified in the phylum *Nucleocytoviricota* [2,3]. This group comprises large nucleocytoplasmic DNA viruses, with particle and/or genome sizes above the average in the virosphere, which have their genetic material replicated in the nucleus or cytoplasm of host cells. According to the International Committee on Taxonomy of Viruses (ICTV), this group is currently divided into three classes, five orders, and 15 families, with the family *Phycodnaviridae* (class *Megaviricetes*, order *Algavirales*) being the one that includes the majority of the known giant algal viruses [4,5].

The family *Phycodnaviridae* is composed of viruses that infect different types of algae. The family consists of six genera, historically based on the host range of these viruses, that was later corroborated by genomic data [5,6]. Among the phycodnaviruses, the genus *Chlorovirus* is the most studied and was the first to be established due to the isolation and characterization of Paramecium bursaria Chlorella virus 1 (PBCV-1), a giant algal virus described as capable of infecting chlorella-like cells and forming lysis plaques under laboratory conditions [7,8]. Chloroviruses are found in inland waters and can infect different species of algae in the family Chlorellaceae that are endosymbionts of protozoa and animals, as far as it is known [9]. There are three well-defined clades of chloroviruses, which we assigned to subgenera and different species based on multiple lines of evidence, including host range, phylogeny, and several genomic analyses [9,10]. The “alphachloroviruses” infect strains of *Chlorella variabilis*; there are currently 68 isolates with sequenced genomes available in the databases [10]. An important feature of these viruses is an apparent host-driven specialization, considering that several species of “alphachloroviruses” can only infect the Syngen 2–3 strain of *C. variabilis* [10]. The “gammachloroviruses” infect *Chlorella heliozoae*. In our analysis of 37 isolates, we demonstrated a great diversity in this group, with the definition of 10 viral species, an open pan-genome, and a curious absence of ecological barriers among these viruses [11]. The “betachloroviruses” are the least studied chlorovirus subgenus, with only 14 isolates described to date. This group has historically been called Pbi-viruses, due to their plaque-forming infection of *Micractinium conductrix* Pbi [12,13].

The genomes of the first betachloroviruses were sequenced almost two decades ago, revealing linear DNA genomes of approximately 320 kbp, more than 300 protein-coding sequences, and up to 10 genes for tRNAs [12]. A few years later, more genomes were sequenced from isolates obtained from different locations, such as the USA, Canada, France, Germany, Norway, and the Czech Republic, revealing a group of viruses with apparently lower genetic diversity when compared to other groups of chloroviruses [13]. An exception was an isolate obtained from samples from Nebraska/USA (NE-JV-1) that had genomic characteristics that distinguished it from other chloroviruses (e.g., absence of genomic collinearity and a more basal phylogenetic position compared to other Pbi-viruses) [13,14]. Given this scenario, isolating and characterizing new betachloroviruses is necessary to improve our understanding of the diversity and biology of these viruses.

In this work, we report the isolation and genomic characterization of 11 new betachloroviruses obtained from one lake and two rivers in Nebraska/USA. Together with the genomic data of the other betachloroviruses, we performed a series of genomic and phylogenetic analyses that led us to propose the existence of 3 species of betachloroviruses and to evaluate the intraspecific diversity of this group. One of the species contains only two isolates, NE-JV-1 and a new, closely related isolate, suggesting that the genetic diversity of this group of viruses is still far from being completely elucidated. From genotypic and phenotypic analyses, we suggest the existence of chlorovirus genomovars, something that can be explored for other giant viruses in the future, opening doors to new discoveries related to the biology of viruses that have been explored limitedly until now.

## 2. Materials and Methods

### 2.1. Virus Isolation and Purification

Water samples were collected from distinct lakes at the Crescent Lake National Wildlife Refuge in Nebraska (USA), as well as nearby rivers, to investigate the presence of chloroviruses between 2017 and 2020 under US Fish and Wildlife Service Approved Permit # CRL-8_002. The samples were filtered through a 0.45 µm membrane (PES filters, Sartorius, Göttingen, Germany) and plaque-assayed using *Micractinium conductrix* Pbi cells grown on FES agar plates [15], incubated for up to two weeks at 25 °C under constant light. Plaques were picked and plaque assayed serially to achieve pure, single plaque isolates. Viral production and purification on sucrose density gradients were performed as previously described [16,17].

### 2.2. Plaque Phenotype Assays

To investigate the plaque phenotypes of different isolates, we used purified viral isolates in plaque assays [7,16]. Each viral isolate that was selected for this project went through at least three rounds of single plaque isolation in series (independent infections). Briefly, 0.1 mL of virus lysate, diluted 10^−2^, 10^−4^, and 10^−6^ in virus stabilization buffer, was mixed with 0.3 mL of concentrated cells (2.0 × 10^8^ cells/mL) and added to 2.5 mL of 0.75% FES-agar medium, which had been diluted by the addition of 0.9 mL of virus stabilization buffer. The top agar mixture was poured onto 1.5% FES-agar plates and allowed to solidify. The infections were all done on the same day with the same cell density, agar thickness and incubation conditions (medium, light, temperature) to allow proper comparison. Plates were incubated at 25 °C in continuous light and evaluated at 5, 7, 12, and 14 days of incubation. Photographs were also taken at these time points. Photographs were made using a Nikon D5200 DSLR camera with a Micro-Nikkor 55 mm 1:2.8 manual focus lens (Nikon Corporation, Tokyo, Japan). Camera settings: shutter speed 1/125, ISO 100, f5.6, custom white balance set to light source. A fluorescent light box with black paper mask was used to backlight plates with the camera mounted to a fixed arm above the light box to provide a consistent scale for comparison of plaques.

### 2.3. Genome Sequencing and Assembly

This study utilized genome sequences from 25 *Micractinium conductrix*-infecting viruses. The genome sequences of 14 betachloroviruses isolated before 2008 were downloaded from public databases [13]. The viral genomes of the other 11 isolates were sequenced and assembled as described by Carvalho et al. [10]. Briefly, new isolates were sequenced using PacBio Sequel II platform (Pacific Biosciences of California, Inc., Menlo Park, CA, USA. Raw long-read data were assembled *de novo* using Canu version 2.2 [18]. All genomes were assembled into a single scaffold with >40× coverage. The new viral genomes were deposited in the NCBI GenBank database with accession numbers PV288763 to PV288773.

### 2.4. Gene Prediction and Annotation

Coding sequences (CDS) were predicted using GeneMarkS version 4.28 [19] online software using default parameters. tRNA predictions were made using ARAGORN version 1.2.41 [20] and tRNAscan-SE [21,22]. Results were processed by removing predicted CDSs shorter than 40 amino acids or fewer than 150 nucleotides from strand ends. CDSs overlapping tRNA regions or containing stop codons were also excluded. Ambiguous tRNAs predicted by both software were removed from the final dataset, as well as pseudo tRNAs predicted by either software. The CDSs were annotated using a combined strategy of software to improve the quality of gene annotation. The NCBI nr database was queried using the Diamond algorithm [23], considering only results with an E-value lower than 10^−5^. Protein domains were identified using HHpred online [24], with results accepted only if the probability was ≥80%. Both results were compared to determine the best final hit. In cases of discordance, InterProScan 5 software [25] was used. For functional characterization, final hits were classified based on the *Nucleocytoviricota* orthologous groups of genes (NCVOG) database [26].

### 2.5. Phylogenetic Analyses

Alignments of amino acid sequences of the NCLDV hallmark genes [3], including DNA polymerase family B, DNA topoisomerase II, SNF2-like helicase, packing ATPase A32, transcription initiation factor IIB, and poxvirus late transcription factor VLTF3, were concatenated to construct a phylogenetic tree of *M. conductrix*-infecting viruses. In addition to the 25 betachloroviruses (Appendix A), Acanthocystis turfacea Chlorella virus 1 (ATCV-1) (GenBank accession: NC_008724.1) was included as the outgroup. The six datasets were aligned using Muscle version 5 [27] and concatenated in MEGA version 11 [28]. The alignments were analyzed using the maximum likelihood method and best evolutionary model defined using ModelFinder implemented in IQ-TREE 2 [29,30], with 1000 bootstrap replicates for statistical support of the nodes. The concatenated and individual trees were visualized using iToL version 7 [31].

### 2.6. Genomic Collinearity Analysis and Species Demarcation Criteria

In order to evaluate the synteny and the alignment of genomic elements, the Mauve version 2.4.0 [32] and the Dynamic Genomic Alignment server (DiGAlign) version 2.0 were employed [33]. The average nucleotide identity (ANI) of the 25 whole genomes was calculated using FastANI version 1.3 [34], hosted on the European Galaxy server (usegalaxy.eu). The average amino acid identity (AAI) was calculated using EzAAI [35]. The results were organized into a similarity matrix that served as input for constructing a heatmap using in-house scripts described previously [10]. Based on the heatmap, the species of betachloroviruses were demarcated using a cutoff of 94% identity, a value consistent with our previous work on alphachloroviruses to define a chlorovirus species [10]. Hierarchical clustering was conducted using Pearson correlation above 0.7 using the hclust package version 3.6.2 in R [36]. For comparison with species from other groups, such as alphachloroviruses and gammachloroviruses, we included ANI values comparing isolates from the species *Chlorovirus vanettense* and *Chlorovirus heliozoae*. The isolates belonging to each chlorovirus species were defined in previous studies, describing the type isolates from the alpha- and gammachloroviruses [10,11]. Hierarchical clustering was conducted as described above for betachloroviruses.

### 2.7. Ortholog Clustering and Pan-Genome Construction

Predicted CDSs were clustered into orthologous groups (COGs) using the OrthoFinder tool version 2.5.5 [37], with an MCL inflation parameter of 4. Each OrthoFinder run included sequential groups of multi-fasta files as input, containing amino acid sequences. In constructing the pan-genome of *Micractinium conductrix*-infecting viruses, both the number of core genes (those shared by all isolates) and total COGs were considered. Results from each run were organized and plotted using GraphPad Prism 9, generating a pan-genome evolution graph. To visualize the overall COG-sharing among the *M. conductrix*-infecting viruses, we utilized Gephi version 0.9 [38] and employed the Force-Atlas 2 algorithm to generate the network layout, with minimal manual adjustments to the nodes to highlight the singletons. To compare the COG-sharing between virus species, we used COG data shared between isolates of each of the three species to build an upset plot using UpSetR version 1.4.0 in conjunction with the ComplexHeatmap package version 2.22.0 in R [39,40].

## 3. Results

### 3.1. Phylogenetics and Genomic Features of New Betachloroviruses

Eleven new chloroviruses were isolated and sequenced in this study, increasing the total number of viruses associated with *Micractinium conductrix* Pbi to 25 isolates (Appendix A). These viruses have genome sizes ranging from 295 kbp (NE-P-2-m) to 374 kbp (NE-P-6-s), with an average GC content of 45%, varying between 44.1% and 46.8%. Betachloroviruses encode hundreds of CDSs, with an average of 377 CDSs per genome, ranging from 337 (NE-P-2-m) to 421 (NE-P-6-s). There is a positive correlation between genome size and the number of CDSs. However, this relationship does not extend to tRNA-coding genes. On average, betachloroviruses encode eight tRNA genes per genome, with observed values ranging from five (NE-JV-1) to twelve (several isolates). Interestingly, the isolate with the smallest genome (NE-P-2-m) is among those with the highest number of tRNAs, while the isolate with the fewest tRNAs (NE-JV-1) has a genome size close to the average for betachloroviruses, approximately 326 kbp (Figure 1).

To reconstruct the phylogeny of betachloroviruses, we used six of the most conserved genes among giant viruses [3], employing isolate ATCV-1 (Gammachlorovirus) as an outgroup. The resulting phylogeny reveals two well-supported clades within betachloroviruses. One of these clades forms a long branch composed of isolates NE-JV-1 and NE-P-6-s (Figure 1 and Appendix A), suggesting substantial genomic divergence. The other clade features much shorter branches, indicative of a greater genetic conservation among its members. We identified a distinct subclade comprising 11 isolates, eight of which were newly obtained from Island Lake in the Crescent Lake National Wildlife Refuge in Nebraska, and from the North and South Platte rivers, also in Nebraska. Remarkably, the remaining three isolates in this group were collected from geographically distant locations: one from Oregon, USA (OR0704.2.2), one from France (Fr5L), and one from the Czech Republic (CZ-2) (Figure 1; Appendix A). This observation suggests the existence of a genetically cohesive group of betachloroviruses with a broad geographic distribution.

In all isolates, we identified conserved genes typical of the phylum *Nucleocytoviricota*, including DNA polymerase family B (PolB), Major Capsid Protein (MCP), Transcription Factor IIB (TFIIIB), DNA Topoisomerase II (TopoII), DNA packaging ATPase (A32), Poxvirus Late Transcription Factor 3 (VLTF3), and DEAD/SFN2-like helicase (SFII). As previously reported for chloroviruses [9,13,41], no homologs of RNA polymerase were found (Figure 2A). Betachloroviruses possess several genes associated with DNA replication, recombination, and repair, such as methyltransferases (specific for adenine and cytosine), primase, and endonucleases. Genes involved in carbohydrate metabolism are also present, including those encoding alginate lyase, chitinase, β-glucanase, and multiple glycosyltransferases. However, the absence of some of these enzymes in certain isolates suggests variability in glycosylation patterns among betachloroviruses (Figure 2A). In addition, a variety of genes involved in other metabolic functions were identified, such as those related to nucleotide metabolism (e.g., dCMP deaminase), membrane transport (e.g., potassium channel protein Kcv), gene transcription (e.g., mRNA capping enzyme), and protein metabolism (e.g., polypeptide chain elongation factor 3) (Figure 2A). We also observed several homologs of capsid proteins previously described in the PBCV-1 genome [42].

One of the most interesting genes found in 23 of the 25 betachloroviruses codes for a homolog of heliorhodopsin proteins (Figure 2A). The two betachloroviruses (NE-JV-1 and NE-P-6-s) that lack this gene are those distantly related to the others in phylogenetic reconstructions. None of the alpha- and gammachlorovirus members have this gene. Some of the betachlorovirus heliorhodopsin-like proteins are about 120 amino acids longer than authentic heliorhodopsin proteins, although they have internal Met residues that could be the translation starting point for the proteins, which would make them similar in size. Two of the 23 betachloroviruses encoding heliorhodopsin-like proteins have truncated versions of the protein (P-NE-12 and NE-P-8-f), with only 109 amino acid residues. All three types of light using rhodopsins have lysine in the seventh transmembrane helix which is essential for the protein to be functional [43]. However, the 23 betachloroviruses have an asparagine in this position.

Chloroviruses are known to encode tRNAs, typically clustered in a central region of the genome, which may supplement the host’s tRNA pool during infection [44]. In betachloroviruses, we similarly observed that tRNA genes are predominantly located near the center of the genome. These viruses encode tRNAs for 10 of the 20 canonical amino acids. Among these, tRNA-Arg, tRNA-Leu, and tRNA-Thr are present in all 25 isolates (Figure 2B). For tRNA-Leu and tRNA-Thr, two anticodons were detected in different isolates, but unevenly distributed. In contrast, tRNA-Asp was found exclusively in the isolate NE-P-6-s. Some tRNAs, such as tRNA-Asn, tRNA-Lys and tRNA-Gly, were present in almost all isolates, with the exception of NE-JV-1, the isolate with the lowest number of tRNAs among the genus *Chlorovirus* (Figure 2B). Interestingly, only tRNA-Asn has duplications in different isolates, with the isolate CZ-2 having four copies of this gene. tRNA-Tyr is also present in most isolates, except for AP110A and the two most divergent isolates, NE-JV-1 and NE-P-6-s. tRNA-Ile and tRNA-Phe show a more heterogeneous distribution, with no clear pattern of conservation among isolates (Figure 2B).

### 3.2. Genomic Organization of Different Species of Betachloroviruses

Chloroviruses have been divided into three main groups based on the host they infect [9,13]. This classification is supported by phylogenetic and genomic data (e.g., genome synteny and gene identity), which reveal significant divergence among the clades [13,41]. When comparing the genomic structure of the newly isolated betachloroviruses with previously described isolates, we observed a high degree of genomic collinearity, indicating strong synteny among these viral genomes (Figure 3 and Appendix A). In some cases, genomic inversions were observed in the first half of the genome (e.g., NE-P-1-L vs. NE-P-3-s). Synteny analysis revealed that the isolates NE-JV-1 and NE-P-6-s have a high degree of conservation in their genomes when compared to each other, yet exhibit a completely different gene organization compared to the other betachloroviruses. This observation supports the distinct long branch these isolates occupy in the phylogeny (Figure 1 and Figure 3). In these two isolates, a genomic inversion was also detected, occurring in the central region of the genome, between 150 and 250 kbp. When comparing the synteny of the new betachlorovirus isolates with representatives from other chlorovirus groups, we observed a complete lack of synteny, clearly underscoring the genomic distinction between the three clades within the *Chlorovirus* genus (Appendix A).

Average Nucleotide Identity (ANI) analysis, combined with hierarchical clustering using Pearson correlation, revealed a clear separation of betachloroviruses into three distinct clusters, corroborating both the phylogenetic and synteny data (Figure 4). Each cluster was defined by isolates sharing at least 94% nucleotide identity (Appendix A). Between clusters, identity values dropped to as low as 82.97%. Based on the >94% ANI threshold, along with supporting evidence from phylogeny and genomic organization, it is possible to delineate three distinct species within the group of betachloroviruses. Furthermore, the ANI values were highly consistent with the results from the Average Amino Acid Identity (AAI) analysis, which also revealed the separation of isolates into three clusters, with intraspecific similarity values ranging from 93.24% to 99.97% (Appendix A).

Among the three clusters—hereafter defined as species I (12 isolates), species II (11 isolates), and species III (2 isolates)—the latter is clearly the most divergent, sharing approximately 83% nucleotide identity with isolates from the other two species. This divergence is supported by the long branch in the phylogenetic analysis. Species I exhibits the highest intraspecific genetic variation, with ANI values ranging from approximately 94% to almost 100%. In contrast, species II shows a narrower range of genetic variability, with ANI values between 96.31% and 99.99%. Species III, composed of only two isolates, displays a genetic difference of only 1.65% between them (Appendix A). Some isolates are nearly identical, showing minimal differences in genetic content and forming tight subclusters with over 99% ANI (Figure 4; Appendix A). This pattern is observed in a subcluster within species I (isolates CVR-1 and CVA-1) and in two subclusters within species II: subcluster 1 (isolates NE-P-8-f, NE-P-3-s, NE-P-4-fs, and P-NE-12) and subcluster 2 (isolates P-NE-13, P-NE-11, NE-P-1-L, and NE-P-1-m) (Appendix A).

To date, betachloroviruses exhibit the lowest genetic diversity within the *Chlorovirus* genus, being composed of only three species with markedly lower intraspecific variation when compared to alphachloroviruses and gammachloroviruses. It is noteworthy that, across all groups, the use of ANI values > 94%, in combination with phylogenomic analyses, has proven to be a robust criterion for species delineation within the *Chlorovirus* genus [10]. For comparative purposes, we analyzed the genetic variation within representative species of the other *Chlorovirus* clades. In *Chlorovirus vanettense* (22 isolates infecting *Chlorella variabilis* NC64A), ANI values range from 94% to 99.98%, with four subclusters identified in which isolates share at least 99% ANI. The most prominent subcluster includes isolates PBCV-1, XZ-4A, XZ-3A, 40-NE-4, 40-NE-3, and 41-NE-5 (Appendix A). For *Chlorovirus heliozoae* (18 isolates infecting *Chlorella heliozoae* SAG 3.83), ANI values range from 94% to 99.16%, with only three subclusters, each composed of two isolates sharing over 99% ANI (Appendix A). These findings reinforce that gammachloroviruses exhibit the highest genetic diversity known among chloroviruses to date [11].

### 3.3. Pan-Genome Evolution and COG-Sharing Pattern of Betachloroviruses

Although betachloroviruses exhibit lower genetic diversity compared to other chlorovirus groups, we still observed an upward trend in their pan-genome, indicating an open pan-genome (Figure 5A). Among the 25 isolates analyzed, a total of 9431 genes were grouped into 678 clusters of orthologous genes (COGs), of which 223 were shared by all isolates, constituting the core genome of betachloroviruses (Figure 5B,C). A total of 334 COGs were shared by 2 to 24 isolates, forming the satellite genome of the group, while 121 COGs were classified as singletons, being present in only one isolate (Figure 5B,C). Among these, isolate NE-P-6-s stands out for containing the highest number of singletons among betachloroviruses, with 13 unique COGs. In contrast, some isolates, such as NE-P-3-s and NE-P-4-fs, have only one singleton each. This observation corroborates the ANI data, as these isolates present ANI values above 99%, and thus, a large number of unique genes would not be expected. Nonetheless, it is noteworthy that even nearly identical genomes harbor exclusive genes, which may reflect biological differences between the isolates.

Species II harbors the largest number of COGs, with a total of 527 shared among the 11 isolates comprising the species, followed by species I with 458 COGs and species III with 382 COGs (Figure 5B,C). When analyzing COG sharing across species, we observed that 285 COGs (42.2%) are shared by at least one isolate from each species, while 262 COGs (38.9%) are exclusive to one or more species. Species III has the fewest exclusive COGs (64; 9.5%), but the highest number of singletons by isolate (Figure 5B,C). Species I and II share 92 COGs, the highest number among interspecies comparisons. In contrast, species III shares the fewest COGs with the others—only 11 with species I and 22 with species II—highlighting the high genetic divergence of species III relative to the others (Figure 5B,C). Among the COGs exclusive to species III, isolate NE-JV-1 contains 8 singletons, and NE-P-6-s has 13. Most of these encode hypothetical proteins, and a substantial portion are classified as ORFans (n = 10/13; 76.9%).

### 3.4. Quasi-Identical Isolates Exhibit Different Plaque Phenotypes

Among the 25 betachlorovirus isolates, we observed some isolates that have nearly identical genomes, i.e., ANI values above 99.5%, some sharing up to 99.99% average nucleotide identity (Figure 6A). In genomes with an average size of 330 kbp, as is the case with these betachloroviruses, this represents a genetic difference of less than 200 nucleotides between some isolates (e.g., CVA-1 and CVR-1; ANI = 99.95%). In species I, there are two nearly identical genomes, which we can consider as genetic variants, or genomovars [45], which share 99.95% of nucleotides. In species II, we observed two groups of isolates that form genomovars, whose isolates share average ANI values of 99.94% (genomovar II.1; isolates P-NE-11, P-NE-13, NE-P-1-L, and NE-P-2-m) and 99.79% (genomovar II.2; isolates P-NE-12, NE-P-3-s, NE-P-8-f, and NE-P-4-fs) (Figure 6A). Interestingly, although some isolates have high genomic identity, they exhibit considerably different plaque morphologies.

Viral isolates of genomovar I.1 (isolates CVA-1 and CVR-1) share 99.95% average identity and the plaque phenotype is highly similar, exhibiting large plaques (>4 mm) with regular edges and clear plaque interiors (Figure 6B,C). It is possible that these isolates exhibit very similar characteristics in general, being almost clones, especially considering that they were isolated from the same region (Appendix A). However, when we analyzed the genomovars of species II, we observed a curious variety of plaque forms in isolates that share more than 99.50% nucleotide identity. The genomovar II.1 isolates exhibit similar plaque phenotypes, sharing ANI > 99.79%. All four isolates form large plaques with regular edges, with NE-P-2-m forming slightly smaller plaques compared to the others. Curiously, isolate P-NE-11 forms large plaques with more diffuse boundaries than the other three isolates in its genomovar; Some plaques of P-NE-11 even display a cloudy ring near the edge that is not seen in others (Figure 6B,C). However, these isolates exhibit only 0.01% genomic divergence from each other, corresponding to a difference of less than 50 nucleotides on average. It is even more intriguing when we look at the genomovar II.2 isolates. These isolates share less than 0.05% genomic divergence, but have highly different plaque phenotypes. Isolates P-NE-12, NE-P-3-s, and NE-P-4-fs form small plaques (<2 mm) with regular edges, while isolate NE-P-8-f forms medium-size plaques (~2.5 mm), with cloudy interior and irregular boundaries. These isolates have ANI = 99.55% (Figure 6). One feature that should not be overlooked is that all isolates show some degree of variability in plaque appearance, with some being more consistent (e.g., CVA-1) and some showing different plaque types (e.g., NE-P-8-f); the plaque representations in Figure 6B are the average plaque appearance observed for each isolate.

It is still unclear what is behind this morphotypic difference in nearly genetically identical viruses. All isolates in Species II genomovar 1 are from Island Lake. Isolates in Species II genomovar 2 are from Island Lake and the North Platte River (NE-P-8-f). Most are from the same sample year. These data suggest that there is a rich biological diversity to be explored in these locations, even considering viruses with very similar genetic characteristics.

## 4. Discussion

Chloroviruses were discovered in the early 1980s and expanded the frontiers of virology by having large icosahedral particles (~190 nm) and large genomes encoding hundreds of proteins, resulting in the proposal of the term “giant virus” [7,8,46,47]. Since then, much has been discovered, especially in relation to the prototype chlorovirus PBCV-1, now classified as part of the species *Chlorovirus vanettense*. Based on many studies involving PBCV-1, we now know that chloroviruses are much more complex than initially imagined. Chloroviruses are pseudoicosahedral particles with capsids composed of many proteins, including at least 5 MCP variants and more than 25 minor capsid proteins [40], and a unique and autonomous glycosylation system [48,49,50,51,52]. Also, they are capable of manipulating the nucleotide metabolism of the host to better suit their molecular requirements during replication [53]. With the discovery of new isolates genetically close to PBCV-1, we observed great diversity and the existence of two main groups of these viruses, with one of them being capable of infecting only one type of host, the so-called OSy-virus, which apparently is derived from a subpopulation of the now-called alphachloroviruses [10,16]. In addition to these, other viruses related to another model virus, ATCV-1, were isolated and characterized, revealing great genetic diversity within the gammachloroviruses [11,54]. However, a group that has not been explored until now is the betachloroviruses (traditionally known as Pbi-viruses). In this study, we sought to fill some gaps in our understanding of their diversity and genomics.

The first betachlorovirus genomes were published in 2007, belonging to isolates FR483 (site of collection: France) and MT325 (site of collection: USA), revealing genomes of 321 and 314 kbp, respectively, and low collinearity compared to PBCV-1 and ATCV-1 [12]. At the time, they were the smallest chlorovirus genomes known. Subsequently, other chloroviruses associated with *Micractinium conductrix* were isolated and sequenced, most obtained from European samples, revealing genomes ranging from 303 to 330 kbp; an analysis comparing these with other isolates’ genomes demonstrated an open pan-genome for the genus *Chlorovirus* [13,41]. In this work, we describe 11 new isolates, increasing the total number of sequenced betachloroviruses to 25, with genomes ranging from 295 to 374 kbp, expanding the limits of genome size for these viruses. These viruses have an average of 370 CDSs, of which a large number (~40%) encode proteins with no known function. This pattern is similar to that observed for other giant viruses, not only chloroviruses, indicating that there is still much to be explored by applying the omics sciences to the study of *Nucleocytoviricota* [2]. Curiously, betachloroviruses are the only chloroviruses that harbor a heliorhodopsin-like homolog, except for two isolates (species III). These genes have been described in coccolithoviruses and are characterized as a light-activated proton transporter that has been hypothesized to be involved in overcoming host defense mechanisms [55]. The role of heliorhodopsin-like proteins in betachloroviruses is still unclear. We speculate that they may be involved in response to light availability during infection. Bornemann and Follman (1997) reported the appearance of concentric rings within viral plaques when betachlorovirus-infected cells were incubated with a circadian change in light intensity, but plaques remained uniformly clear under constant light [56]. Assessing the impact of heliorhodopsins under different light conditions may provide valuable insights into the role of these proteins in chloroviruses.

Synteny analyses revealed that betachloroviruses generally exhibit strong genomic collinearity among themselves, with a few inverted regions in some isolates. Notably, two viruses stand out for having a gene organization markedly distinct from the others. These results reinforce findings from previous studies, which have highlighted the extensive genomic diversity among the three chlorovirus groups [41]. Using ANI and AAI analyses and applying a 94% identity threshold among isolates, we were able to define three distinct species of betachloroviruses. This threshold aligns well with observations in alphachloroviruses and gammachloroviruses [10,11], supporting its robustness as a criterion for species delineation within the *Chlorovirus* genus. When integrated with additional evidence such as phylogenetic relationships and host range, this approach strengthens taxonomic proposals in line with ICTV guidelines [57].

Although betachloroviruses display lower genetic variability compared to other chloroviruses, they still exhibit an expanding pan-genome. This suggests that the continued isolation and characterization of new strains may uncover genetic novelties and further increase the slope of the pan-genome curve. As observed in alphachloroviruses and gammachloroviruses [10,11], the core-genome curve reaches a plateau, indicating a probable set of essential genes that define members of this viral group. It is important to highlight that one of the identified betachlorovirus species is currently represented by only two isolates. The recent discovery of NE-P-6-s is particularly significant, demonstrating the profound impact of virus prospecting efforts on our understanding of viral diversity. Until now, isolate NE-JV-1 had been regarded as a phylogenetic outlier due to its considerable divergence from other betachloroviruses [9,13,14]. Depending on the molecular marker used for phylogenetic reconstructions, NE-JV-1 occasionally appeared more closely related to gammachloroviruses [14,41], underscoring how limited sampling can skew evolutionary interpretations.

Among the betachlorovirus isolates, we found no clear relationship between geographic origin and genetic diversity, unlike what has been considered for other giant virus groups [58]. Viruses isolated from distant countries such as France, Norway, and Canada exhibit high genomic similarity and belong to the same species. Conversely, isolates from a single water sample in the United States show greater genetic divergence and are classified into different species (e.g., both P-NE-9 and NE-P-8-f are from the same North Platte River sample collected in October, 2017). It is noteworthy that all currently known betachlorovirus isolates originate from samples collected in North America and Europe. Expanding virus isolation efforts to underrepresented regions of the world will be essential to gain a more complete understanding of the true diversity and global distribution of these viruses.

When analyzing intraspecific diversity, we observed only slight genomic variation among some isolates, with several pairs displaying ANI values above 99%. A comprehensive study involving over 1000 viral genomes, predominantly bacteriophages but also including eukaryotic viruses, proposed the use of ANI to define intraspecific viral units, particularly when identity exceeds 99% [45]. The authors introduced the term “genomovar” to describe each of these subunits within a viral species. Our data support the existence of genomovars in chloroviruses. Remarkably, we observed that some of these genomovars exhibit distinct phenotypic traits despite having nearly identical genomes. One of the most striking phenotypic differences is the variation in plaque morphology, a characteristic often used to infer viral virulence and to screen for genetically modified clones [59,60,61]. Curiously, even in highly purified isolates, we can still observe slightly different plaque phenotypes, indicating a considerable heterogeneity in biological traits of chloroviruses. It is important to note that plaque assays for comparison purposes should be performed simultaneously to control for potential variability that may arise between experiments. Plaque assays performed at different times with different cell cultures may result in a difference in plaque appearance, making data interpretation difficult. Despite our efforts to purify viral plaques, we still observed variations in plaque sizes. When analyzing the raw genome sequences, we found no evidence of different viruses in the samples. We still do not have an explanation for the observed variation in plaque size among chlorovirus isolates, and this can be better evaluated in future studies.

It is possible that subtle genetic differences, such as small deletions or point mutations, underlie these morphotypic variations. Similar phenomena have been documented in other giant viruses, such as poxviruses. For instance, vaccinia virus has been shown to display at least two distinct plaque morphologies, associated with minor variations in specific genes, although a definitive genotype-to-phenotype link has yet to be established [62,63]. This pattern is also observed in RNA viruses like enteroviruses and flaviviruses, where single-nucleotide polymorphisms may influence plaque morphotype and pathogenic potential [64,65]. Further investigation into betachlorovirus genomovars, especially through integrative genomic and biological analyses, will be key to uncovering the molecular basis of intraspecific variation and advancing our understanding of the complex biology of these giant algal viruses.

## Figures and Tables

**Figure 1 viruses-17-01096-f001:**
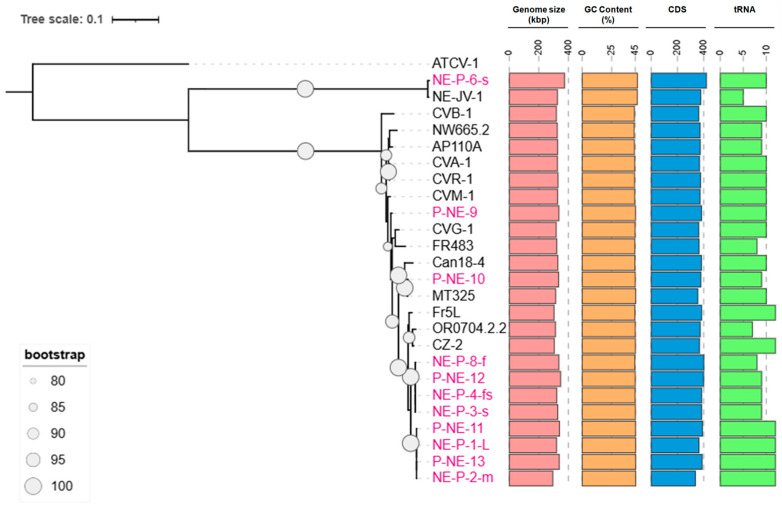
Phylogenetics and genomic characteristics of betachloroviruses. Phylogenetic reconstruction of betachloroviruses. The phylogeny was made based on six concatenated genes (A32, PolB, SFII, TFIIB, Topo, VLTF3) considered the best hallmark genes for the phylum *Nucleocytoviricota*. New isolates are represented in pink labels, while already known isolates are in black. ATCV-1, a gammachlorovirus, was used as an outgroup. Only bootstrap values above 80 are represented. The tree scale refers to the substitution rate of amino acids. Colored bars indicate the different genomic elements found in the virus isolates.

**Figure 2 viruses-17-01096-f002:**
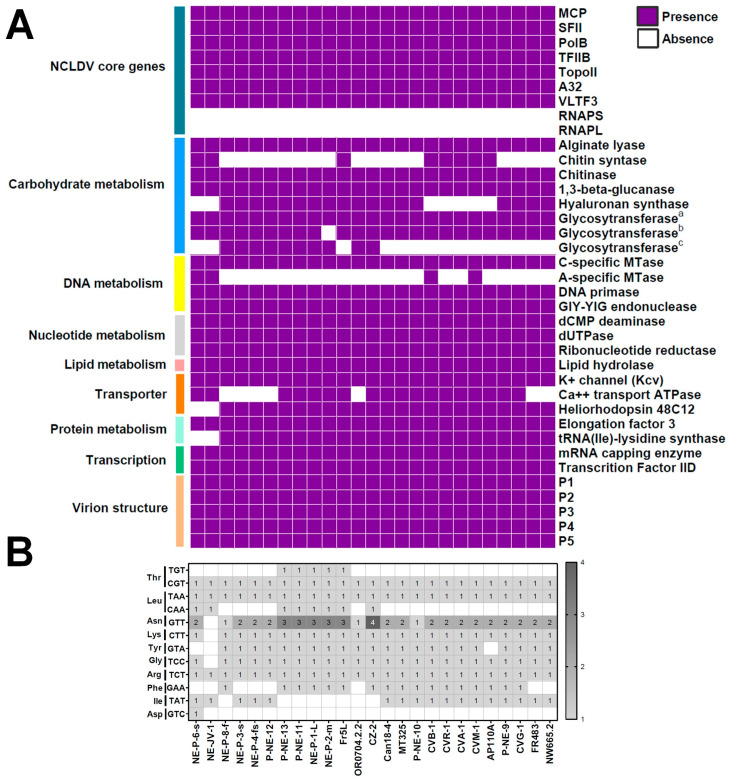
Genomic content of betachloroviruses. (**A**) Genes commonly found in betachloroviruses, separated based on functional categories. Only five of the 24 known minor capsid proteins of chloroviruses (P1 to P5) are represented in the graph, although homologs of other genes have been found in different isolates. Three different glycosyltransferase homologs were included to highlight possible distinct glycosylation machinery/pattern expressed by each isolate; (**B**) tRNA diversity in betachloroviruses. The number of anticodons is indicated in the respective square of each isolate. The scale bar indicates the range of tRNA counts in the isolate.

**Figure 3 viruses-17-01096-f003:**
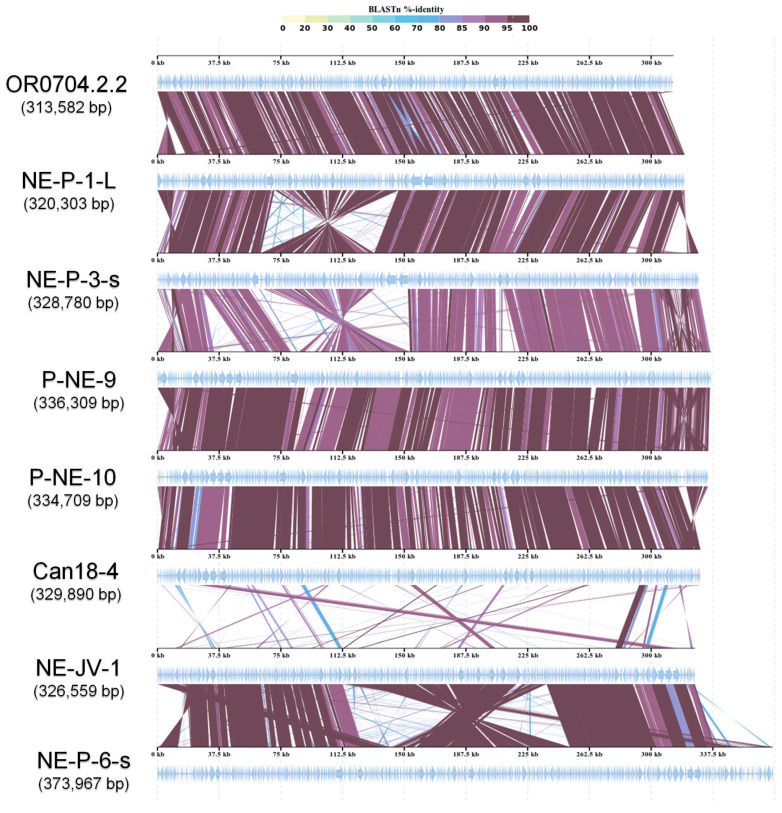
Genome synteny of betachloroviruses. New and old isolates are included in the analyses, indicating conserved collinearity among the genomes, except the NE-JV-1 and NE-P-6-s. The isolates of each cluster were chosen randomly for this analysis. Inversions in some genomic regions are observed. The color grade represents BLASTn identity percentage.

**Figure 4 viruses-17-01096-f004:**
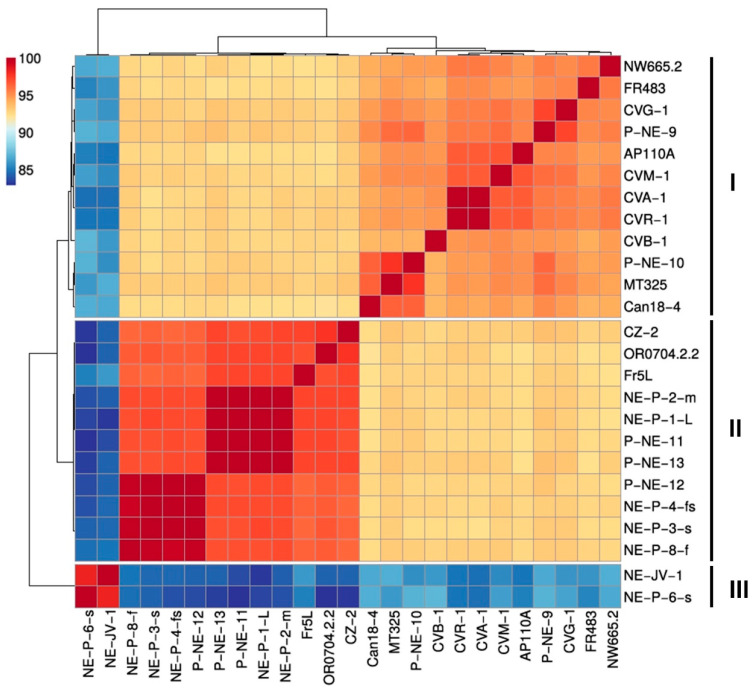
Betachloroviruses are clustered into three species. Hierarchical cluster analysis reveals three distinctive groups in the ‘Betachlorovirus’ subgenus. The grouping reflects a 94% cutoff value for species designation. We refer to these groups as species I, II and III. The scale bar indicates the range of ANI values.

**Figure 5 viruses-17-01096-f005:**
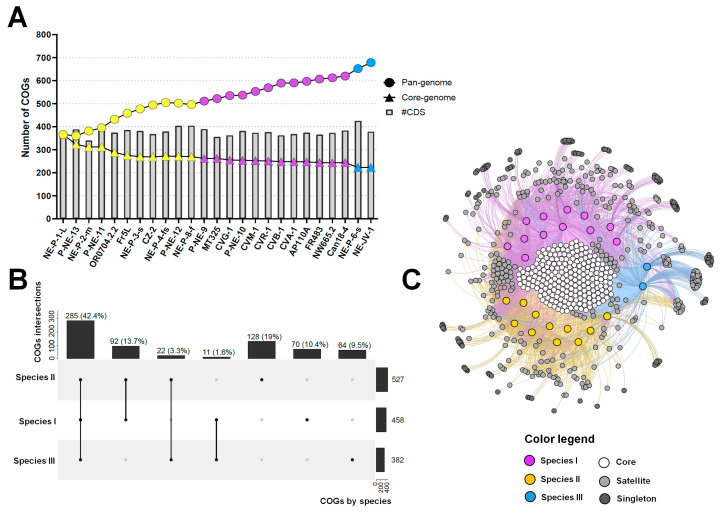
Pan-genome and COG-sharing pattern in betachloroviruses. (**A**) Pan-genome evolution of betachloroviruses. Pan-genome and core-genome are represented by circles and triangles, respectively; (**B**) Upset plot of COG-sharing considering the three betachloroviruses species. The number of COGs found per species is indicated in the horizontal bars. Vertical bars indicate the number of COGs shared between the species; (**C**) Bipartite network graph representing the COG-sharing pattern of betachloroviruses. Colored nodes represent the isolates, while nodes in the grayscale represent the COGs. White nodes are the core genome, gray nodes represent the satellite genome, and dark gray nodes stand for singletons. In all images, each species is represented by a different color: species I, purple; species II, yellow; species III, blue.

**Figure 6 viruses-17-01096-f006:**
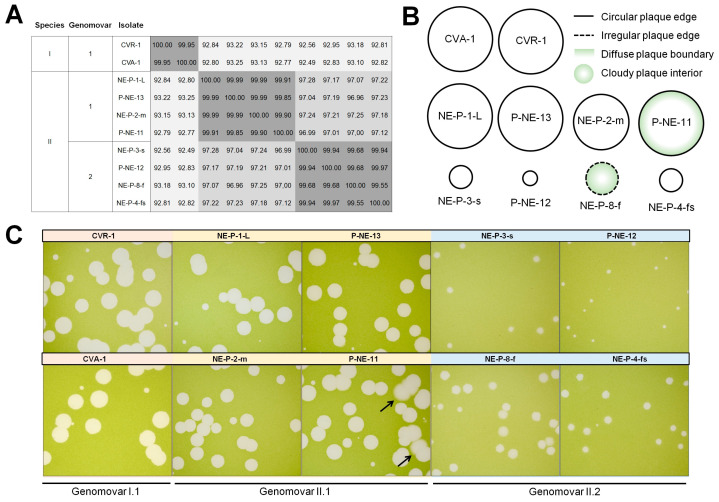
Betachloroviruses genomovars exhibit different plaque phenotypes. (**A**) ANI matrix heatmap of betachloroviruses sharing ANI >99%, grouped by species and genomovars; (**B**) Schematic representation of different plaque phenotypes of betachloroviruses. Plaques can be classified by size as small (<2 mm), medium (2–4 mm), and large (>4 mm), with regular (continuous line) or irregular (discontinuous line) edges, and the clarity of plaque morphology, being regular (white plaque) or cloudy (green plaque). Plaques also exhibit varying levels in the clarity of the plaque field, being clear (white plaque) or cloudy (green plaque), as well as differences in the definition at the plaque boundary, having a sharp plaque boundary (white inside border) or a soft plaque boundary (green inside boarder); (**C**) Plaque phenotypes of different isolates of betachloroviruses. All photos were taken after 12 days of infection. Black arrows indicate soft plaque boundaries in isolate P-NE-11.

## Data Availability

All genomic data used in this work is publicly available at the GenBank database. The new viral genomes were deposited in the NCBI GenBank database with accession numbers PV288763 to PV288773. All genomic data used in this research is publicly available at the BioProject PRJNA1154233 “Chlorovirus Diversity”. This site contains links to all chlorovirus genomes in GenBank as well as the raw sequence data in the SRA, and information about the isolates.

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
