# Peer review of "New Isolates of Betachloroviruses Shed Light on the Diversity and Biological Complexity of an Unexplored Group of Giant Algal Viruses"

_viruses, 2025, doi:10.3390/v17081096_

Round 1
Reviewer 1 Report
Comments and Suggestions for Authors
This study reports 11 new betachloroviruses infecting Micractinium conductrix, which increases the total number of sequenced isolates to 25. Genomic and phylogenetic analyses reveal three distinct species within this group, one of which includes two highly divergent genomes. Interestingly, despite high genomic similarity among some isolates, they display markedly different plaque morphologies. The results are important and will contribute towards our understanding of fine-scale genomic heterogeniety of viruses infecting same genus and species of green alga, and how that relates to their infection success and other phenotypes.
I found the study to be well-written, with results described in great details. I do not have any concern regarding the methods or results. However, I was a bit unclear about the choice of ANI. Why did the authors choose 94% ANI? This needs to be clarified further.
Mohammad Moniruzzaman
Author Response
Comments 1: This study reports 11 new betachloroviruses infecting Micractinium conductrix, which increases the total number of sequenced isolates to 25. Genomic and phylogenetic analyses reveal three distinct species within this group, one of which includes two highly divergent genomes. Interestingly, despite high genomic similarity among some isolates, they display markedly different plaque morphologies. The results are important and will contribute towards our understanding of fine-scale genomic heterogeniety of viruses infecting same genus and species of green alga, and how that relates to their infection success and other phenotypes.
I found the study to be well-written, with results described in great details. I do not have any concern regarding the methods or results. However, I was a bit unclear about the choice of ANI. Why did the authors choose 94% ANI? This needs to be clarified further.
Response 1: Dear Mohammad Moniruzzaman, thank you very much for your kind and encouraging comments. The choice of 94% ANI was initially based on our previous work on alphachloroviruses, where we established this as one of the criteria for defining a chlorovirus species (see Carvalho et al., 2024 – Ref. 10 in this manuscript). In addition, our data on betachloroviruses presented in this manuscript corroborate this criterion, since using phylogeny and 94% ANI, we confidently separated the isolates into species, in this case, revealing three betachlorovirus species. We have included this information in the revised version of the manuscript for clarity.
Reviewer 2 Report
Comments and Suggestions for Authors
This study presents a genomic and phenotypic analysis of 11 newly isolated betachloroviruses, significantly expanding our understanding of this understudied group within the Phycodnaviridae family. The work is well-structured, methodologically robust, and provides valuable insights into viral diversity, genomovar classification, and genotype-phenotype relationships. The findings are novel and contribute meaningfully to the field of giant virus research.
Major concern: Plaque phenotype analysis and interpretation
The manuscript reports intriguing differences in plaque morphology among genomovars with near-identical genomes (ANI >99.5%), suggesting a potential link between minor genetic variations and phenotypic outcomes. However, several critical issues need to be addressed to strengthen the validity and interpretation of these findings.
- The study should clarify whether plaque assays were performed in biological replicates (e.g., independent infections) and/or technical replicates (e.g., multiple plates per isolate). Variability in plaque size/morphology can arise from subtle differences in host cell density, agar thickness, or incubation conditions.
- Plaque phenotypes were assessed at 12 days post-infection. Were earlier/later timepoints evaluated? Delayed lysis or host recovery could influence observations.
- Differences in Microctinium conductrix Pbi cell batches (e.g., growth rate, metabolic state) may contribute to phenotypic variability. Were host cells standardized across assays?
- Typically, a single viral isolate produces plaques of relatively uniform size, though minor variations (e.g., up to two-fold differences) may occasionally occur. However, as shown in Figure 6C, many isolates (particularly genomovar I.1 and II.1) exhibit plaque diameter differences of up to ten-fold on the same plate. This substantial variation raises the possibility of viral population heterogeneity, suggesting that the isolates may not be as clonally pure as initially assumed. To confirm this, I recommend performing plaque-PCR on individual plaques to assess genetic consistency.
Minor Points
Line 98: Line 98: Please correct “Gottingen” to “Göttingen” (standard spelling for the German city).
Figures 1 and S1: Several bootstrap values are misaligned; please ensure they are positioned adjacent to their corresponding clade nodes.
Figure 3 and Figure S3: Replace “nt” with “bp” for consistency with dsDNA genomic nomenclature.
Line 571 Abbreviations: Please change “Coding DNA Sequence” to “Coding Sequence” (CDS) and update “Kilobase pairs” to “Kilobase pair” (kbp) (singular form for the unit).
Author Response
Comments 1: This study presents a genomic and phenotypic analysis of 11 newly isolated betachloroviruses, significantly expanding our understanding of this understudied group within the Phycodnaviridae family. The work is well-structured, methodologically robust, and provides valuable insights into viral diversity, genomovar classification, and genotype-phenotype relationships. The findings are novel and contribute meaningfully to the field of giant virus research.
Response 1: Dear reviewer, thank you very much for your valuable and constructive comments on our manuscript. Please, find below a point-by-point answer to each of your concerns.
Comments 2: Major concern: Plaque phenotype analysis and interpretation
The manuscript reports intriguing differences in plaque morphology among genomovars with near-identical genomes (ANI >99.5%), suggesting a potential link between minor genetic variations and phenotypic outcomes. However, several critical issues need to be addressed to strengthen the validity and interpretation of these findings.
The study should clarify whether plaque assays were performed in biological replicates (e.g., independent infections) and/or technical replicates (e.g., multiple plates per isolate). Variability in plaque size/morphology can arise from subtle differences in host cell density, agar thickness, or incubation conditions.
Response 2: Each viral isolate that was selected for this project went through at least 3 rounds of single plaque isolation in series (independent infections). For the results presented in Figure 6 for comparative purposes, the infections were all done on the same day with the same cell density and same agar thickness and same incubation conditions (medium, light, temperature). That is, the isolates were evaluated in parallel. This information was added to the methodology section for clarity. Please, see revised topic 2.2.
Comments 3: Plaque phenotypes were assessed at 12 days post-infection. Were earlier/later timepoints evaluated? Delayed lysis or host recovery could influence observations.
Response 3: As stated in the Materials and Methods, line 112-114, photographs were taken at 5, 7, 12, and 14 days. We did not observe significant differences between the days evaluated, and we chose day 12 to show because of a better contrast to observed the lysis plaques.
Comments 4: Differences in Microctinium conductrix Pbi cell batches (e.g., growth rate, metabolic state) may contribute to phenotypic variability. Were host cells standardized across assays?
Response 4: Yes, the host cells have been standardized across assays. The Micractinium conductrix Pbi cells have been used in our lab for nearly 40 years. These cells are stored as agar slants that are renewed every 6 months and stored at 4 °C in low light. The slant cultures are used to renew the liquid starter cultures. The starter cells are renewed every 10 days. They are ready to use when they reach near stationary phase (approximately 8 x 108 cell/mL).
Starter cells are used to inoculate liquid medium for the “daily cells” (which are the cells used for experimentation). The rationale for using starter cells in the stationary growth phase to establish new daily cell cultures is to ensure synchronized growth.
The daily cells are ready to use in experiments when they reach a cell density of 1 – 1.5 x 107 cells/mL (approximately mid-log phase) for the desired day of the experiment. The culture medium, lighting condition,s and room temperature are standard. Evidence of the long-term stability of these cells is the reproducibility of the plaque morphologies of viral isolates over many years (in some cases, decades).
Comments 5: Typically, a single viral isolate produces plaques of relatively uniform size, though minor variations (e.g., up to two-fold differences) may occasionally occur. However, as shown in Figure 6C, many isolates (particularly genomovar I.1 and II.1) exhibit plaque diameter differences of up to ten-fold on the same plate. This substantial variation raises the possibility of viral population heterogeneity, suggesting that the isolates may not be as clonally pure as initially assumed. To confirm this, I recommend performing plaque-PCR on individual plaques to assess genetic consistency.
Response 5: We recognize that some variations in the plaque size occurred in our assays, despite all our efforts to purify the plaques. In this context, the viral isolates used for all the genomic sequencing projects went through at least 3 rounds of single plaque isolation. The phenomenon of getting two (or more) plaque morphologies is not uncommon for the chloroviruses. We do not have a confirmed explanation for this phenomenon.
The suggestion of using a PCR amplification with subsequent DNA sequencing of the amplicons is a doable objective but it is unclear as to which gene/DNA sequencing region should be targeted. Years ago, when one of us (I.V.A.) was preparing the Alphachlorovirus KS1B for sequencing, she observed two plaque morphologies. She took both types and did a PCR-based sequencing of the DNA polymerase gene using degenerate primers. Both plaque types gave identical sequences. In this case, the preferred method is to sequence the entire genome of each isolate. Functionally, we have done this.
Checking the raw sequences, we found no evidence of different viruses. Taking all this into account, we still don't have an explanation for the observed variation in plaque size among chlorovirus isolates. This information was added to the discussion section of the manuscript.
Comments 6: Minor Points - Line 98: Line 98: Please correct “Gottingen” to “Göttingen” (standard spelling for the German city).
Response 6: Done.
Comments 7: Figures 1 and S1: Several bootstrap values are misaligned; please ensure they are positioned adjacent to their corresponding clade nodes.
Response 7: Done.
Comments 8: Figure 3 and Figure S3: Replace “nt” with “bp” for consistency with dsDNA genomic nomenclature.
Response 8: Done.
Comments 9: Line 571 Abbreviations: Please change “Coding DNA Sequence” to “Coding Sequence” (CDS) and update “Kilobase pairs” to “Kilobase pair” (kbp) (singular form for the unit).
Response 9: Done.
Round 2
Reviewer 2 Report
Comments and Suggestions for Authors
The authors have addressed all necessary revisions, and I have no remaining comments.